# Biomedical Relevance of Novel Anticancer Peptides in the Sensitive Treatment of Cancer

**DOI:** 10.3390/biom11081120

**Published:** 2021-07-29

**Authors:** Olalekan Olanrewaju Bakare, Arun Gokul, Ruomou Wu, Lee-Ann Niekerk, Ashwil Klein, Marshall Keyster

**Affiliations:** 1Environmental Biotechnology Laboratory, Department of Biotechnology, University of the Western Cape, Bellville 7535, South Africa; 3056605@myuwc.ac.za(R.W.); 3255882@myuwc.ac.za (L.-A.N.); 2Department of Plant Sciences, Qwaqwa Campus, University of the Free State, Phuthaditjhaba 9866, South Africa; Gokula@ufs.ac.za; 3Plant Omics Laboratory, Department of Biotechnology, University of the Western Cape, Bellville 7535, South Africa; aklein@uwc.ac.za

**Keywords:** anticancer peptides, apoptosis, cancer, cytolysis, host-defense peptides, peptide delivery

## Abstract

The global increase in cancer mortality and economic losses necessitates the cautious quest for therapeutic agents with compensatory advantages over conventional therapies. Anticancer peptides (ACPs) are a subset of host defense peptides, also known as antimicrobial peptides, which have emerged as therapeutic and diagnostic candidates due to several compensatory advantages over the non-specificity of the current treatment regimens. This review aimed to highlight the ravaging incidence of cancer, the use of ACPs in cancer treatment with their mechanisms, ACP discovery and delivery methods, and the limitations for their use. This would create awareness for identifying more ACPs with better specificity, accuracy and sensitivity towards the disease. It would also promote their efficacious utilization in biotechnology, medical sciences and molecular biology to ease the severity of the disease and enable the patients living with these conditions to develop an accommodating lifestyle.

## 1. Introduction

Cancer is used synonymously to mean either malignant tumors or neoplasms and refers to a group of diseases that affect any part of the human body [1]. According to the World Health Organization (WHO), it is the leading cause of global mortality, accounting for about 20 million deaths in 2020, in which the most common causes of death include lung (1.80 million), colon and rectum (935,000), liver (830,000), stomach (769,000) and breast (685,000) [2]. It occurs through the rapid formation of abnormal cells that develop uncontrollably to invade the surrounding body parts and organs through the process of metastasis, the primary cause of death from cancer [3]. Cancer affects all age groups, but the incidence of cancer rises with age due to the build-up risk of specific cancers that increases with age, coupled with the fact that the capacity for cellular amelioration mechanisms becomes less effective with age [4].

The causes of cancer have been linked to a series of interactions between an individual’s genetic factors and the three categories of external agents [5]. These external agents include biological carcinogens through infections from some parasites, viruses and bacteria [6]; physical carcinogens through the interaction with ultraviolet and ionizing radiation [7]; and chemical carcinogens through exposure to asbestos, tobacco smoke, water contaminants, such as arsenic, and food contaminants such as aflatoxins [8]. Approximately thirteen percent of cancers diagnosed in 2018, for instance, were caused by carcinogenic infections such as hepatitis B virus, hepatitis C virus, Epstein Barr virus, human papillomavirus (HPV) and Helicobacter pylori [9]. Particularly, the risk of liver and cervical cancer increases with some HPV and Hepatitis B and C viruses, while the incidence of HIV increases the risk of cervical cancer substantially [10].

Understanding the molecular mechanism of cancer formation from uncontrolled cell division to tissue invasiveness can shed light on the mutations of genes and proteins involved in cell cycle inactivation and suppression [11]. Epigenetic alterations have also been reported in cancer, where some protein-coding genes were altered due to methylation in colon cancer [12]. The epigenetic alterations implicated in DNA-repair genes that cause reduced expression of DNA-repair proteins have been involved in cancer progression’s genetic instability at the early stage [13]. Another cause of mutation in cancer is DNA-mismatch repair or homologous recombinational repair (HRR) in defective cells 9899. DNA repair inhibition has also been said to be involved in heavy metal-induced carcinogenicity. At the same time, the regulatory activities of miRNAs can target and reduce the expression of some protein-coding genes [14].

Appropriate and efficacious treatment is associated with early and correct cancer diagnosis because every cancer treatment is related to a specific treatment regimen. The current treatment regimens of cancer include immunotherapy, hormone therapy, stem cell transplant, biomarker testing (the use of genes, proteins and other substances referred to as tumor markers or biomarkers), radiotherapy, chemotherapy and surgery [15]. The goal of these treatment procedures is primarily to either remove/kill cancer from the body (primary treatment), to reduce the chance that cancer will recur by killing the remaining cancer cells after primary treatment (adjuvant treatment) or to relieve the side effects of treatment (palliative treatment). Despite these interventions, late side effects are associated with these treatment regimens, which depend on the type of cancer being treated and where in the body the cancer treatment is conducted [16]. Certain general side effects reported include lymphedema, fertility issues, nerve problems, sexual health issues, urinary issues, insomnia, anemia, loss of appetite, thrombocytopenia, constipation, delirium, diarrhea, edema, fatigue and others. Researchers and the international community have called for a more sensitive treatment regimen with little or no side effects to reduce cancer incidence and its growing menace. The use of anticancer peptides (ACPs) would eliminate the limitations of most cancer treatments, such as low solubility and restrictive and negative side effects [17].

Host defense peptides (HDPs), also known as host-defense antimicrobial peptides (AMPs), are key components of the innate immune system in all life forms, both vertebrates and invertebrates. In fact, some invertebrates, such as insects and crustaceans, do not have an adaptive immune system and use only the innate immune system for their protective mechanism. HDPs have emerged as amphipathic and short cationic biomolecules of diverse sequences, synthesized from the cells and tissues of complex life forms [18]. HDPs perform various functions, which include antimicrobial, anti-inflammatory, immunomodulatory, antioxidant, protease inhibitors, antiparasitic, and anticancer (the use of anticancer peptides (ACPs)) functions [19]. Many HDPs have been identified, which differ in structure and sequence and have been classified into α- helical peptides, β- sheets with disulfide bridges, cyclic peptides, and peptides with extended flexible loop structures [20]. Several HDP databases exist as a catalog for over 2600 naturally occurring anticancer peptides (ACPs) [21]. However, synthetic ACPs for cancer treatment bring consistency and stability and allow for more consistent and accurate research results tailored to providing the desired effect by binding to a specific receptor without interfering with other receptor subtypes under proper usage conditions with fewer side effects and great benefits [22]. This review attempts to explore the biological importance of anticancer peptides (ACPs) in the treatment of cancer through their mechanism of action and technologies used for their identification and delivery, as well as their challenges.

## 2. Modes of Action of Host-Defense Peptides (HDPs)

HDPs have many modes of action, which seem to be conserved to some degree across different cell types, which include bacteria and cancer cells [23]. Cationic HDPs can interact with the membrane of cancer cells as they predominantly contain negatively charged molecules [24]. An example of such peptides is Leucine-leucine-37 (LL-37), which has membranolytic activities (Table 1). Such cationic HDPs have an electrostatic interaction with cancer cell membranes, which plays an important role in eliciting a cytotoxic effect on the cancer cells [25,26]. Arias et al., 2020 [22] undertook a study to improve the anticancer activity of synthetic HDPs. Arias et al., 2020 [22] could substitute an arginine for a lysine amino acid, which resulted in an enhanced electrostatic interaction and selectivity for Jurkat Leukemia cells compared to non-cancerous peripheral blood mononuclear cells.

It was observed that HDPs have the ability to interact with the cell membrane of cells, such as bacteria, and neutralize the charge allowing the HDPs to further penetrate through the membrane, thus increasing their cytotoxic effects [27]. It should be noted that cancer cells have an abnormal cell membrane composition when compared to normal cells [28]. The fluidity of cell membranes may be determined by the cholesterol concentration and distribution throughout the membrane. Cancer cells with lower cholesterol deposition will have an increased sensitivity to certain HDPs [24,29]. A study by Frislev et al., 2017 [30] observed the use of a liprotide HAMLET (human α-lactalbumin made lethal to tumor cells) in order to kill cancer cells (Figure 1).

The study showed that the liprotides were able to increase the fluidity of the membrane. The same study also observed that by knocking down Annexin A6, a protein that is responsible for plasma membrane repair, they could enhance the liprotide’s killing effects. A study by Mamusa et al., 2017 [31] observed that derivatives of the HAL-2 peptide were able to directly damage yeast cell membranes resulting in increased death of the cells. The same study also observed that, by replacing a methionine with a valine amino acid, the therapeutic effect was decreased to all cells tested. These studies show that the composition of the antimicrobial peptide is important and that making informed modifications may result in enhanced therapeutic effects in different cell types such as bacterial or cancer cells.

Antimicrobial peptides have also been reported to interact with cell receptors in order to either engage secondary effector proteins or initiate immunomodulatory processes such as the inflammatory signaling pathways [32,33]. An example of such anticancer peptide with immunomodulatory activities includes high mobility group box protein 1 (HMGB1) (Table 1) [34]. It should be noted that the selective interaction of these HDPs is attributed to their physicochemical properties [32]. It has also been observed that identifying the exact functions of HDPs is difficult as their expression levels may dictate the role they play in processes such as inflammation (pro-inflammatory or anti-inflammatory) [32,35,36]. A study by Li et al., 2019 [37] showed that the antimicrobial peptide LL-37 interacted with the P2X_7_ receptor, which allowed for the internalization of the HDP into the cell. Once the AMP LL-37 activated the P2X_7_ receptor, it induced pore formation in the membrane [38]. It was observed that a scrambled form of LL-37 would not interact strongly with a neutral membrane and cause the pore-forming phenomenon, showing how important the structure of an antimicrobial peptide was to its ability to interact with receptors [37].

Research is ongoing to improve the therapeutic effects of ACPs while reducing their toxicity [38]. In order to achieve the aforementioned goal, studies modifying natural cationic ACPs have been performed. A study by Alexander et al., 2002 [39] performed a study to observe the effects of AMPs to affect macromolecular synthesis. The study showed that Pleurocidin derivatives at low levels were able to inhibit macromolecular synthesis within cells while causing less damage to the cell membrane. This study shows that by understanding the composition of AMPs and how they function, we may produce AMPs that are active and effective at lower concentrations, which may reduce their toxicity.

**Table 1 biomolecules-11-01120-t001:** Mechanism of some HDPs used for the treatment of cancer forms.

N/B	ACPs	Cancer Type	Mechanism	HDP Sequences	References
**1**	Leucine-leucine-37 (LL-37)	human oral squamous cell carcinoma (OSCC) cells	Membranolytic activity towards tumor cell using “toroidal pore” mechanism	LLGDFFRKSKEKIGKEFKRIVQRIKDFLRNLVPRTES	[40]
**2**	Human defensins or α-defensins (human neutrophil peptides (HNP-1, HNP-2, and HNP-3))	human myeloid leukemia cell line (U937), human erythroleukemic cell line (K562), and lymphoblastoid B cells (IM-9 and WIL-2).	cytolytic activity	HNP-1: ACYCRIPACIAGERRYGTCIYQGRLWAFCC HNP-2: CYCRIPACIAGERRYGTCIYQGRLWAFCC HNP-3: DCYCRIPACIAGERRYGTCIYQGRLWAFCC	[41]
**3**	Human β-defensin-3 (hBD3)	HeLa, Jurkat and U937 cancer cell lines	Binding to cell membrane containing phosphatidylinositol 4,5-bisphosphate [PI(4,5)P2] to cause cytolysis	GIINTLQKYYCRVRGGRCAVLSCLPKEEQIGKCSTRGRKCCRRKK	[42]
**4**	Bovine lactoferricin (LfcinB) LTX-315	drug-resistant and drug-sensitive cancer cells	cytolysis and immunogenicity	FKCRRWQWRMKKLGAPSITCVRRAF	[43]
**5**	high mobility group box protein 1 (HMGB1)	All cancer types	immature dendritic cells activation and tumor-specific cytotoxic generation	GRRRRSVQWCAVSQPEATKCFQWQRNMRKVRGPPVSCIKRDSPIQCIQA	[34]
**6**	Gomesin	murine and human cancer cell lines along with melanoma and leukemia	carpet model for destroying the membrane	QCRRLCYKQRCVTYCRGR	[44]
**7**	Mastoparan-C	lung cancer H157, melanocyte MDA-MB-435S, human prostate carcinoma PC-3, human glioblastoma astrocytoma U251MG and human breast cancer MCF-7 cell lines	induce apoptosis and activate phospholipase, inhibition of ATPase activity selectively	INLKALAALAKKIL	[45]
**8**	Cecropin B1	NSCLC cell line	tumor growth inhibition using pore formation and apoptosis	KWKIFKKIEKVGRNIRNGIIKAGPAVAVLGEAKAL	[46]
**9**	Magainin 2	human lung cancer cells A59 and in Ehlrich’s murine ascites cells	formation of pores on cell membranes	GIGKFLHSAKKFGKAFVGEIMNS	[47]
**10**	Bufforin IIb	Leukemia, breast, prostate, and colon cancer	Destruction of the membrane using mitochondrial apoptosis	TRSSRAGLQFPVGRVHRLLRK	[48]
**11**	Brevinin 2R	T-cell leukemia Jurkat, B-cell lymphoma BJAB, colon carcinoma HT29/219 and SW742, fibrosarcoma L929, breast adenocarcinoma MCF-7, and lung carcinoma A549 cells	Lysosomal death pathway (LDP) and autophagy-like cell death	KLKNFAKGVAQSLLNKASCKLSGQC	[37]
***12***	*Limnonectes fujianensis* brevinvin (LFB)	lung cancer H460, melanoma cell, glioblastoma U251MG, colon cancer HCT116 cell lines	penetrate the lipidic bilayer causing cell death	FLPLAVSLAANFLPKLFCKITKKC	[49]
**13**	Phylloseptin-PHa	breast cancer cells MCF-7, breast epithelial cells MCF10A	penetrate the lipidic bilayer causing cell death	FLSLIPAAISAVSALANHF	[50]
**14**	Ranatuerin-2PLx	prostate cancer cell PC-3	cell apoptosis using caspase-3	GIMDTVKNAAKNLAGQLLDKLKCSITAC	[51]
**15**	Dermaseptins (DRS)	prostate cancer cell PC-3	Pore formation and internalization of the lipid bilayer	GLWSKIKEVGKEAAKAAAKAAGKAALGAVSEAV	[52]
**16**	Chrysophsin-1,-2 and-3	human fibrosarcoma HT-1080, histiocytic lymphoma U937, and cervical carcinoma HeLa cell lines	disrupt the plasma membrane	FFGWLIKGAIHAGKAIHGLIHRRRH	[53]
**17**	Ss-arasin	human cervical carcinoma HeLa and colon carcinoma HT-29	cytotoxicity against cancer cells	SPRVRRRYGRPFGGRPFVGGQFGGRPGCVCIRSPCPCANYG	[52]
**18**	Turgencin A and turgencin B	melanoma cancer cells A2058 and the human fibroblast cell line MRC-5	Pore formation and internalization of the lipid bilayer	Turgencin A: GPKTKAACKMACKLATCGKKPGGWKCKLCELGCDAV Turgencin B: GIKEMLCNMACAQTVCKKSGGPLCDTCQAACKALG	[54]
**19**	D-K6L9	breast and prostate cancer cell lines	reduce neovascularization	LKLLKKLLKKLLKLL	[48]
**20**	Dusquetide (SGX942)	neck and head cancer	Binds p62 to cause membrane damage	RIVPA	[37]

## 3. Novel Anticancer Peptides (ACPs) Used in Cancer Therapy

The negative charges exhibited more by cancer cells than by normal cells are contributed by several factors: overexpression of heparan sulfate proteoglycans, the abundance of zwitterionic phosphatidylethanolamine, PS overexpression, glycolipids glycosylation deregulation, and membrane glycoproteins with O-glycosylation repeat [55]. The loss of asymmetry of the phospholipid distribution within the extracellular and intracellular plasma membrane layers in tumor cells exposes the tumor cells’ PS exterior [56]. Internalizing anticancer HDPs at the hydrophobic core also compromises the stiffness and fluidity of the cancer cell membrane and promotes their lytic effects [51].

Several ACPs are available in the databases, such as antimicrobial database APD3 (https://wangapd3.com/database/antiC.php, accessed on 26 June 2021) for therapeutic intervention against various cancer forms. Some of them include Magainin 2, which creates pores during cell membrane damage to establish its anticancer activities against human lung cancer cells, A59, and in Ehlrich’s murine ascites cells [47], Buforin IIb, which disrupts the membrane using mitochondrial apoptosis to establish its anticancer activities against leukemia, breast, prostate and colon cancer [57]. Brevenin-2R causes lysosomal and autophagy-like cell death to T-cell leukemia Jurkat, B-cell lymphoma BJAB, colon carcinoma HT29/219 and SW742, fibrosarcoma L929, breast adenocarcinoma MCF-7, and lung carcinoma A549 cells [49]. *Limnonectes fujianensis* brevinvin penetrates the lipid bilayer causing cell death to lung cancer H460, melanoma cell, glioblastoma U251MG, and colon cancer HCT116 cell lines [49]. Phylloseptin-PHa penetrates the lipid bilayer causing cell death to breast cancer cells MCF-7, breast epithelial cells MCF10A [50]. Ranatuerin-2PLx establishes cell apoptosis using caspase-3 against prostate cancer cell PC-3 [51], and Dermaseptins establishes pore formation of the prostate cancer cell PC-3 [52].

The novelty of the use of ACPs in the sensitive treatment of cancer has been well-established, utilizing their structure and functional relationships. ACPs, such as LL-37—an alpha-helical peptide that belongs to the cathelicidin family—are under clinical trials in a phase I-II stage against melanoma using intratumor injection [40]. A nonamer peptide, LTX-315—derived from structure–activity relationship studies of HDP bovine lactoferricin—has also been tested as an efficacious drug either as a single or in combination with immune checkpoint inhibitors (such as anti-CTLA4/anti-PD-1) for its effect of causing changes in a tumor microenvironment such as an upsurge of T effector cells, necrosis of tumors and a decrease in immunosuppressive cells in a human clinical trial (phase 1) [58]. The phase II stage clinical trial of LTX-315 is ongoing to validate the intratumor administration for patients living with advanced metastatic sarcoma using tumor-infiltrating lymphocytes (TILs). Dusquetide (SGX942), a novel innate defense regulator of both pathogen-associated molecular patterns (PAMPs) and damage-associated molecular patterns (DAMPs) by binding p62, is being explored in phase III clinical trials against neck and head cancer [59]. D-K6L9, in combination with IL-12, was found to reduce neovascularization in breast and prostate cancer cell lines, while (KLAKLAK)2 uses apoptotic-induced-mitochondrial-membrane damage to treat Hela cell lines [60,61]. Some novel ACPs are summarized in Table 1 against different cancer forms as well as the mechanisms involved.

## 4. Mechanism of ACPs for Cancer Treatment

ACPs can exhibit anticancer activity through various mechanisms; mainly through membrane disruption or pore formation. These membrane-active mechanisms are an essential feature of ACPs, as the chance of resistance developing against this type of treatment is low [60]. ACPs can also express anticancer activity through non-membranolytic mechanisms by focusing on intracellular targets, mediating innate host immunity or actively blocking pathways that lead to tumor formation.

The carpet model can describe one of the mechanisms of ACPs, which directly involves cell membrane disruption. This carpet model starts with HDP, which is positively charged and interacts with a negatively charged phospholipid located on the outer layer of the cell membrane. This interaction causes the peptides to align parallel to the cell membrane without protruding into the phospholipid bilayer and, in turn, covering the cell, thus it is termed ‘carpet.’ Once enough peptides protect the cell membrane, a threshold concentration will be reached. Then, the peptides start to rotate on themselves and insert into the membrane, causing permeabilization of the cell membrane. The continuation of this process will ultimately lead to the formation of micelles (Figure 2) [60]. A study by Arias et al. [22] demonstrated the membrane disruption mechanism of ACP Tritrp-Arg by incorporating fluorescent dye propidium iodide (PI) into the treatment of Jurkat cells. This study showed an increase in PI fluorescent intensity, indicating that the cell membrane was permeated, and the fluorescent dye could bind to the DNA within the cell. Specific examples of these ACPs are indicated in Table 1.

In the barrel and stave model (Figure 3), transmembrane channels are formed from the collection of monomer peptides on the cell membrane; the monomer peptides undergo a structural change and aggregate together to form the ‘stave’ within the membrane bilayer. The insertion process is the result of the aggregation, which forces the peptides into the lipid core region; the insertion of the peptide creates a hydrophilic channel that blocks off the hydrophobic part of the bilayer. Once the channel has formed, more monomers will be accumulated to further increase the channel’s size. In addition, the cell membrane is also weakened due to the hydrophobic forces exerted by the peptide. At the moment, the only known ACP that employs this mechanism is alamethicin [43].

The toroidal pore (Figure 4) can be described as temporary holes within the cell membrane created by host defense peptides that are long enough to span across the bilayer before the disintegration of the membrane. The host defense peptides will align parallel to the cell membrane. Once the peptides reach a certain concentration, the peptides will start to insert into the membrane causing the lipid layer to bend inwards, resulting in a toroidal-like pore structure, which is lined by the polar head of the lipid layer and the inserted peptide [60,62]. The toroidal pore allows for host defense peptides to gain access to the intracellular space of the cell, where peptides can disrupt pathways that are responsible for DNA replication, protein synthesis, or permeating the mitochondrial membrane. Examples of host defense peptides that use this mechanism include cecropin A, protegrin-1 and magainin-2 [62].

Host defense peptides capable of translocation across cellular membranes can trigger apoptosis via disruption of the mitochondria. The host defense peptides, which can enter the intracellular space of the cell through the pores, will permeate the membrane of the mitochondria, which will result in the expulsion of protein cytochrome c. The release of cytochrome c will cause a cascade of effects whereby Apaf-1 oligomerization will be activated followed by the activation of caspase 9, which will lead to the conversion of pro-caspase 3 to caspase 3. The caspase 3, in turn, will induce apoptosis [63]. Host defense peptides, such as LL-37 and CATH-2, which has previously been shown to be able to cross the cell membrane and induce apoptosis [64].

Anticancer inhibition can also occur without directly acting on the cancerous cell membrane. Certain host defense peptides can actively compete for binding sites, blocking pathways that lead to cancer formation (Figure 5). We can see this demonstrated using synthetic peptide BLBD, which can bind to β-catenin and LEF-1 sites, which results in the decreased formation of breast cancer cells [65]. In addition, certain pathways require protein to undergo conformational changes, such as dimerization or isomerization, before binding to active sites; host defense peptides can be used to bind and merge with these proteins, ultimately disabling them [66,67].

Peptides can also be used to target tumor cells to make for more efficient drug delivery and to enhance the effects of chemotherapy drugs. This peptide has already been used in Phase I/II clinical trials for advanced solid tumors. For example, the peptide BT1718 will target overexpressed MMP14 sites found in tumor growth and improve drug delivery and efficiency [68]. Another way that the host defense peptides can be used for the treatment of cancer is by mediating the body’s own immune system. Peptides such as alloferon-1 and alloferon-2 found in insect venom can induce natural killer cells in mammals. Cancer treatment with only alloferon-1 has shown cancer suppression activity close to the results from a low dose of chemotherapy [67].

In cancer treatment, peptides can be used in a variety of ways. This includes the use of peptides as medications (for example, as angiogenesis inhibitors), tumor-targeting agents that transport cytotoxic pharmaceuticals and radionuclides (targeted chemotherapy and radiation therapy), hormones and vaccinations [69]. Proapoptotic peptides, for instance DP1, could be used to treat a variety of solid human tumors such as head and neck cancers, melanomas, and papillomas. This primary application of ACPs could be an adjuvant to pre-existing treatment methods such as chemotherapy or radiotherapy [70].

Transient pore formation is another mechanism used by HDPs to establish their effects. Melittin, for instance, can bind on the vesicle translocated and redistributed on both sides of the membrane to induce stable and transient membrane permeabilization above a critical peptide-to-lipid ratio at the nanomolar range that allows for the transmembrane conduction of atomic ions without glucose or larger molecule leakage [71]. HDPs also use electroporation to establish functional domains of intracellular peptides or to gain insight into the peptide inhibition of signal transduction in adherent cells, such as chondrocytes, through transiently forming pores [43]. The use of membrane depolarization is another mechanism utilized by the HDPs to establish their therapeutic effects through the release of cellular contents, leading to death [72]. For instance, abeta-induced membrane depolarization in PC12 cells has been seen to be sensitive to metabotropic glutamate receptor mGluR(1) antagonists and to pertussis and cholera toxins with the involvement of G-protein.

## 5. Discovery Techniques for the Identification of Sensitive HDPs

Several technologies have been explored to identify HDPs of novel importance. Bakare et al., 2020 [20] used the HMMER (a name given by the software developers Sean Eddy and Travis Wheeler) and other in silico technologies to develop putative ACPs that could target the ligand-binding sites of cadherin-1 to monitor the peptides’ prognostic efficacy in patients living with cancer. HMMER is used to identify homologous protein or nucleotide sequences and sequence alignment to discover more sensitive peptides [73]. Grafskaia et al., 2018 [74] used transcriptomic technologies in combination with in silico analysis to discover ten novel synthetic antimicrobial peptides from the sea anemone *Cnipodus japonicas*, three of which were verified to be potent against bacterial strains. Transcriptomic technologies are used to study the sum of an organism’s RNA transcripts, giving rise to the genome’s ability to synthesize biomolecules within the cells and control its gene expression regulation [75].

More potent, cost-effective, broad-spectrum HDPs have been developed by Liu et al., 2020 [76] using advanced computer-assisted design strategies to address challenging problems of translating a primary sequence to peptide structures to solve myriad multi-drug resistance problems. Fields et al., 2020 [77] used a machine learning approach and a simple biophysical trait to develop 20-amino acid bacteriocin peptides that can traverse the membrane of pathogens causing cytotoxic, antimicrobial and hemolytic activities. Antimicrobial peptides with potent and broad-spectrum activities have also been recently designed using molecular engineering technologies to elucidate the peptide motifs and translation opportunities to explore rational design for industrial collaboration [78]. Molecular engineering technologies are being explored in this regard to design and test molecular properties, behavior and interactions in order to assemble better peptides, systems and processes for specified functions [79].

Apart from this, Aruleba et al., 2018 [80] studied the ligand-binding sites and molecular docking interaction of *Slc2a4* as a target for the treatment of cancer with putative HDPs using HMMER for the discovery of the peptides. Wang, 2017 [81] described the discovery, design and treatment strategies of HDPs using the conserved genes from the genomic and proteomic approaches outside the HDP genes with experimental validation, which guarantee a complete mapping utilizing the procedures of sequence shuffling, library screening, hybridization and de novo design. Tucker et al., 2018 [82] used bacterial self-screening of surface-displayed peptide libraries to discover diverse physicochemical parameters of next-generation antimicrobials to unravel the current limitations of peptide applications. Molecular dynamic (MD) approaches have been adopted to study the relationship between the biological function and mechanism of HDPs to optimize these antibiotic candidates. In silico technologies, such as HMMER, with molecular validation techniques have also been used to explore the use of novel HDPs as diagnostic candidates against three bacterial pneumonias and HIV with great promise for industrial collaborations in a lateral flow device [83,84].

## 6. Molecular Validation Techniques Used for HDPs

Several anticancer peptides (ACPs) have been discovered to manage cancer and other diseases, which are being subjected to molecular validation to ascertain their activities and application in terms of sensitivity and specificity. Tincho et al., 2016 [85] used molecular methods, such as cell viability, cytotoxicity and other anti-HIV assays, to show the anti-HIV activities of HDPs against HIV gp120 and NL4-3 receptors. The use of cell viability for peptides’ molecular validation has limitations because cell viability is very diverse due to the redox potential of the cell population, cell membrane integrity, or the activity of cellular enzymes that form a snapshot of cytotoxicity or drug efficacy [86]. Even specially designed cell viability indicators, such as fluorescence microscopes, microplate readers or flow cytometers, have positive and negative attributes with their sensitivity, reliability and compatibility being determined by relevant cell lines [87].

Williams et al., 2017 [84] employed site-directed mutagenesis to generate variant HDPs from previously identified ones and used molecular validation techniques, such as a lateral flow device (LFD) binding assay and nanotechnologies recombinant technologies, to explore their activities against HIV p24. The advantages of the use of LFD include easy pick-up from testing sites and pharmacies, with the fast generation of results within 30 min, thus making it popular [88]. However, researchers compared 5869 people with both an LFD and a PCR test in mass testing in Liverpool. Seventy of these people were positive from the PCR tests and, of these 70, only 28 were positive on LFD tests, showing a sensitivity of about 40% [89]. Cardoso et al., 2021 [78] enumerated the molecular targets and mechanisms involved in several HDPs’ activities and pharmacokinetics. An example of a molecular target that binds one or more signaling peptides or signaling proteins is tropomyosin-receptor kinase B (TrkB), which is bound and activated by the neurotrophic protein brain-derived neurotrophic factor (BDNF) [90]. Prada-Prada et al., 2020 [91] used circular dichroism, cytotoxicity, minimum inhibitory concentration (MIC), growth and time-kill kinetics to explore Ib-M’s mechanisms and structural activities of peptides against *E coli*. Scanning electron microscopy (SEM), transmission electron microscopy [73], NaCl permeability, agarose diffusion, and inhibitory concentration assays were utilized to evaluate the hemolytic activities, toxicity, stability and mechanism of action of chicken hemoglobin HDPs [92]. These technologies, as listed, are necessary to assess and ensure the specific activities of the optimized peptides for clinical trials with negligible toxicity. Regmi et al., 2017 [93] highlighted the combinatorial drug therapy of HDP from *Bacillus amyloliquefaciens* with beta-lactams evaluating its stability, MIC, susceptibility testing, synergy testing and antibiofilm property against pathogens. This combined application of two or more HDPs or the use of antibiotics with HDPs is very promising to prevent the development of antimicrobial resistance and provide the susceptible host with an optimized therapy [94].

## 7. Challenges for the Use of Anticancer Peptides (ACPs) in Cancer Treatment

Many advancements were made in cancer therapy because of the complications experienced with cancerous cells’ resistance to cancer treatments and the low specificity of the currently employed drugs in chemotherapy [95]. The anti-neoplastic drugs, presently used in cancer treatments, consist of various damaging side effects, mainly because they target rapidly dividing cells rather than only cancerous cells [96]. Thus, anticancer peptides (ACPs) provide an alternative anticancer drug [95]. ACPs are beneficial because they differentiate between neoplastic and non-neoplastic cells, which interact specifically with the negatively charged membrane components (these differ between cancer and non-cancer cells [95]. ACPs have a similar ability, in which these peptides have specificity towards malignant cells [96]. Despite the several effective in vivo studies published on ACPs, none of these approaches have made it onto the market [95]. The challenges for using ACPs in cancer treatments are the poor bioavailability of peptides, the toxicity of peptides, immune response to treatments, and the cost-inefficiency of these approaches.

In the cancer treatment serum, proteolytic degradation is a major threat to the potency of the antitumor peptides [96,97], as this promotes unspecific binding to the serum components, reducing the half-life of these molecules in the serum, leading to proteolytic degradation, thus decreasing the bioavailability and affecting the stability of the serum. To overcome these challenges, research needs to be conducted to improve the pharmacodynamic properties of the serum. Baker et al., 1993 [97] successfully exhibited this phenomenon of bioavailability of the peptides by introducing D-amino acids or substituting naturally occurring L-amino acids by diastereomers, such that the peptides’ cytotoxicity was decreased against normal, non-cancerous cells and that these diastereomers peptides maintained their anticancer activities. Ultimately, these researchers recorded a reduction in serum inactivation and enzymatic degradation in in vivo studies [98]. Other approaches were also taken by introducing vector-mediated delivery of the genes that encode the active anticancer peptides [80], using various delivery systems (liposomes, polymer nanoparticles or quantum dots [96]. Some of the naturally occurring ACPs and some synthetic ACPs are more stable in the serum, especially synthetic ACPs containing D-amino acids, which confer stability against proteolytic degradation [96].

All anticancer peptides (ACPs) represent a degree of anticancer activity. However, not all these peptides are suitably selective against cancer cells [96]. In cancer treatments, the hydrophobicity of the peptides is essential for membrane penetration, and a fundamental advocate of the membrane interactions of peptides is amphiphilicity. ACPs promote tumor tissue penetration and therefore kill target cells rapidly by perturbing the integrity of the plasma membrane. The problem arises when peptides showing cancer cell specificity only fall within a narrow range of 0.53–0.78 and have non-tilted helical peptide structures, when these specific peptides, with further analysis, indicate toxic effects to both cancerous and non-cancerous cells and when tilted helical peptides represent a nonspecific means of cell membrane lysis. Therefore, research needs to be conducted that focuses on the structure of peptides, and the activity relationship needs to be further understood to design promising novel antitumor agents [95].

The introduction of foreign ACPs into a host can elicit treatment neutralizing antibodies and/or cause potentially harmful allergic responses in cancer patients [96]. Thus, a few approaches have been considered to overcome this problem. To avoid causing deleterious anti-ACP immune responses, the introduction of host-defense peptides (HDPs) could be a possible alleviation strategy, or the co-administration of foreign ACPs with immunosuppressive drugs. Alternatively, the encapsulation of ACPs in liposomes that are engineered to deliver their cargo straight to the tumor sites could be promising, as this minimizes the opportunity for the host to acquire anti-ACP immunity [96].

In most new treatment approaches, the cost efficiency of the process is always a major setback. In the case of ACPs and AMPs, the cost of isolating naturally occurring ACPs and even the production of synthetic ACPs tends to exceed the costs of the normal production of conventional chemotherapeutic agents that are currently employed in cancer treatments [95,98]. This is not a problem with lipopeptides and other small peptides since chemoenzymatic methods and solution synthesis are employed to generate them; however, when it comes to large peptides, the production costs increase [99].

## 8. Conclusions

The challenges of cancer treatment procedures and associated side effects have necessitated the quest for new therapeutic interventions to combat its menace. ACPs have demonstrated potential therapeutic efficacy against different forms of cancer because of their specificity and the inability of tumors to develop resistance towards them. To this end, this review provides a comprehensive account of novel anticancer ACPs ranging from modes of action, relevant ACPs used in cancer treatment, mechanisms of action, molecular validation, technologies employed in their discovery, and limitations for their use as anticancer agents. It is essential to state that the different structures in terms of amino acid composition and residues gave rise to variation in the mechanisms of action of various ACPs and their targets for cancer therapy. Utilizing the strategies from this review would enhance the development of more sensitive and specific HDPs to solve cancer incidence completely.

## Figures and Tables

**Figure 1 biomolecules-11-01120-f001:**
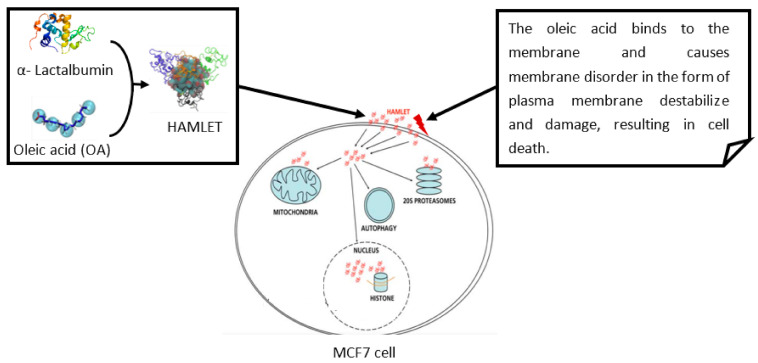
Adapted from Frisley et al., 2017 [30] which represents the liprotide complex, HAMLET, and the target organelles in MCF7 (human breast adenocarcinoma cell line) cells. The HAMLET complex is comprised of α-lactalbumin and oleic acid. The α-lactalbumin function in the complex is to retain the oleic acid (OA) in solution and transport the OA to vesicles of MCF7 cells, increasing membrane fluidity (Frislev et al., 2017). Accessed at https://s100.copyright.com/CustomerAdmin/PLF.jsp?ref=1054733c-0e16-42b8-9270-647b8f875da8 (accessed on 19 June 2021).

**Figure 2 biomolecules-11-01120-f002:**
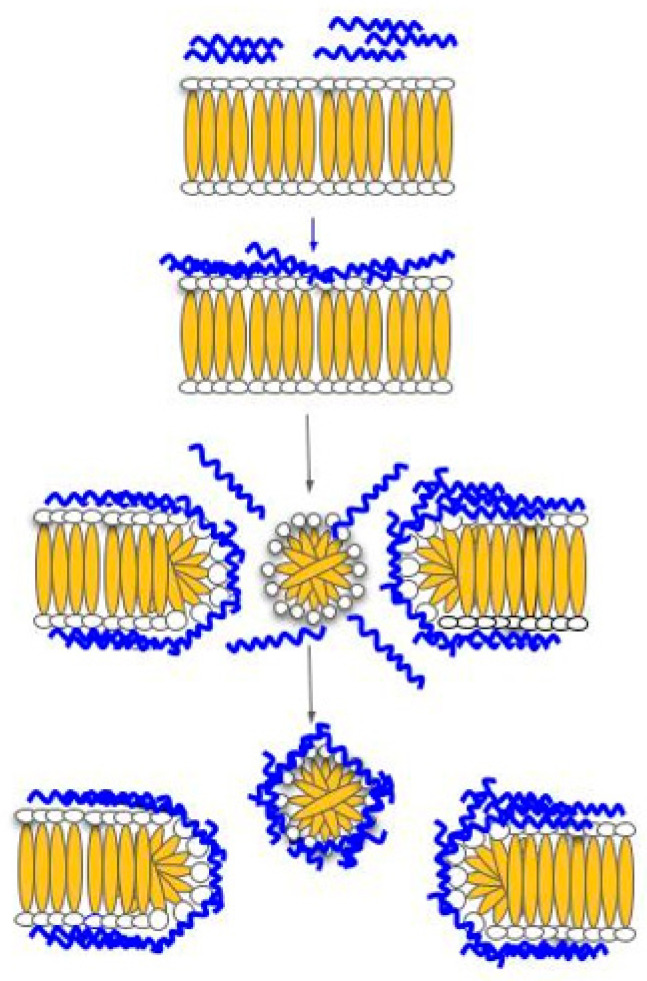
Carpet model. In this model, HDP can disrupt the membrane by orienting parallel to the surface of the lipid bilayer and forming a carpet like layer, in which membrane permeation will occur.

**Figure 3 biomolecules-11-01120-f003:**
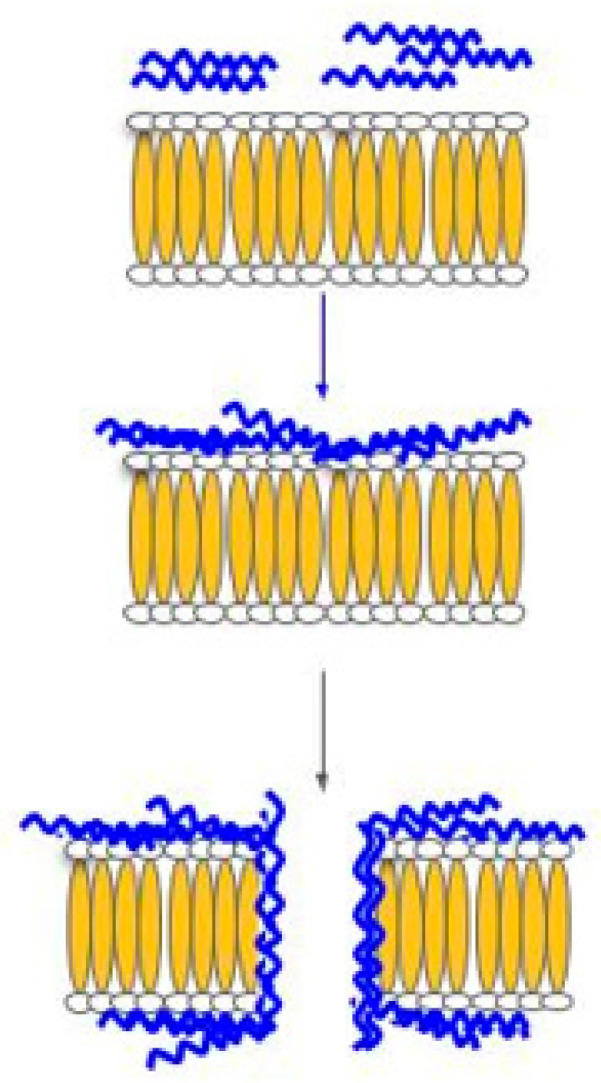
Barrel and stave model. In this model, HDP attaches to the cellular membrane and aggregates. HDP is then inserted into the membrane bilayer so that the hydrophobic peptide regions align with the lipid core region, and the hydrophilic peptide regions form the interior region of the pore.

**Figure 4 biomolecules-11-01120-f004:**
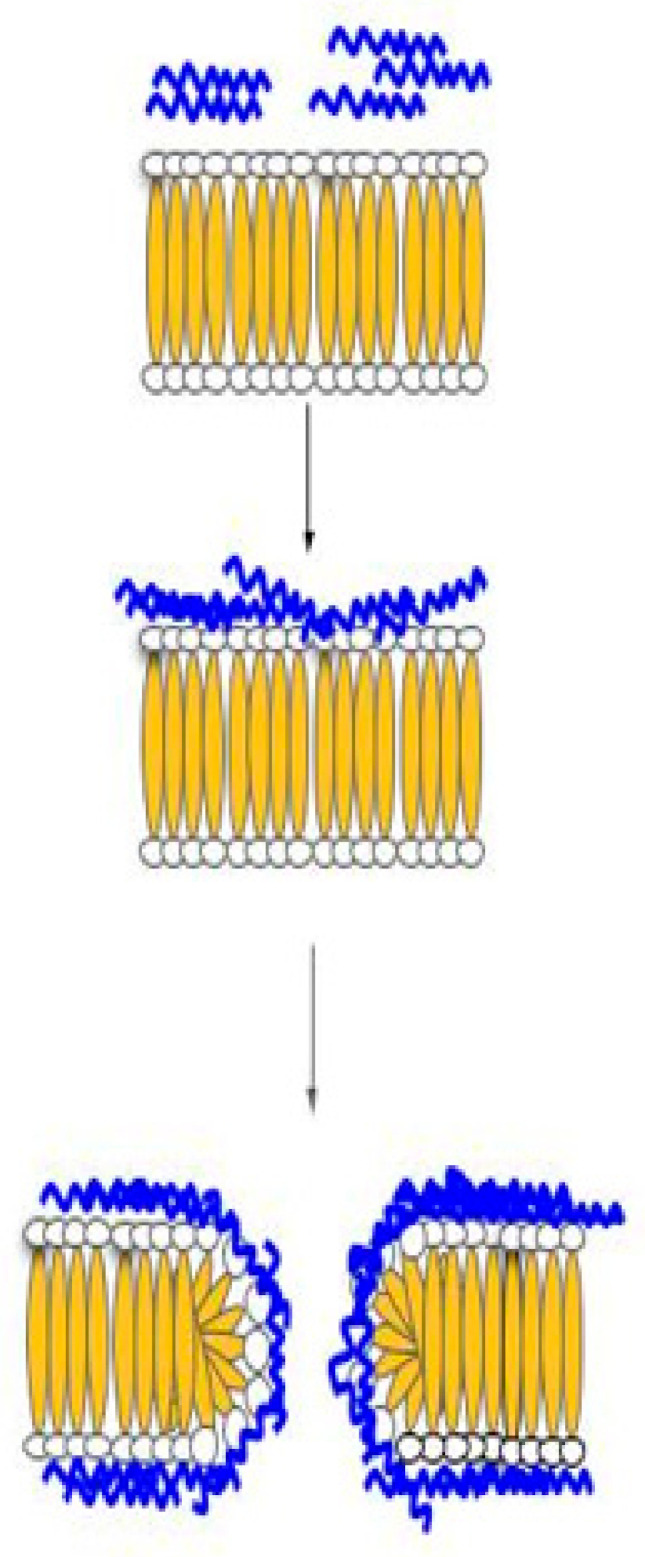
Toroidal pore model. Prior to membrane permeation, HDP, which is aggregated on the membrane surface, will cause the single layer lipid to bend inwards so that the pore is lined by both inserted HDP and the lipid head of the membrane.

**Figure 5 biomolecules-11-01120-f005:**
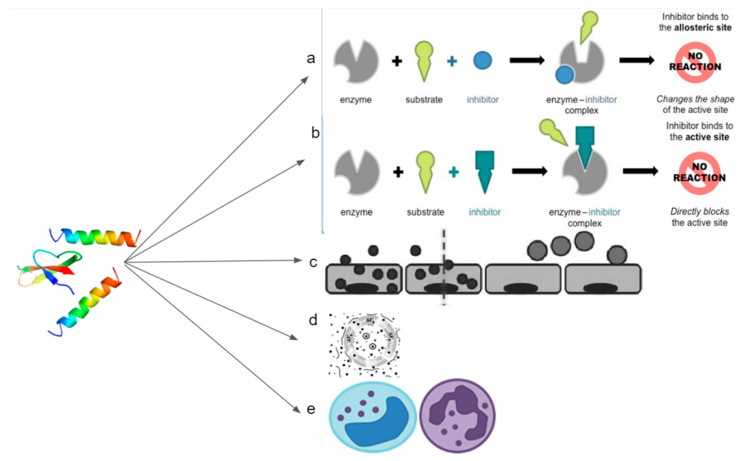
Image showing that the host defense peptide can act on tumors/cancer through various mechanisms. (**a**) Competitively bind to the sites of precursor proteins and thus preventing downstream interaction; (**b**) Simulate the conformation regulation domain of the target protein to inhibit its conformation dependent activation; (**c**) Target cancer cells to improve efficacy of chemotherapy drugs; (**d**) Cancerous cell membrane disruption; (**e**) Mediate immunity.

## Data Availability

Not applicable.

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
