# Peer review of "Biomedical Relevance of Novel Anticancer Peptides in the Sensitive Treatment of Cancer"

_biomolecules, 2021, doi:10.3390/biom11081120_

Round 1
Reviewer 1 Report
The article "Identification and biological relevance of new anticancer peptides (ACPs) in the sensitive treatment of cancer", which reviews about ACPs, taking into account the mechanisms of action of these molecules, as well as validation and identification techniques. The topic of the article is quite interesting and can be better worked on in a future review, being of great interest to the field of anticancer peptides. In its present form, the article looks more like a collection of information than a review article. I believe that the article can bring a fruitful discussion in this field, being better written and discussed, better exploring the information and works cited. In general, the information that can be valuable for the field related to peptides and can be accepted for publication in Biomolecules after major revisions. However, I have some criticisms on attached file.

Author Response
Thank you for your comments. I hereby upload the answers to the comments.
Many thanks.

Reviewer 2 Report
In this review, O. Bakare et al. have introduced development of the anticancer peptides (ACPs) in cancer therapy. On the whole, the article cited kinds of reference, which would be quite helpful for readers to understand the potation of ACPs in the specifical and precise program for cancer therapy. However, major revision is necessary before this article is accepted for publication on the “biomolecules”.
1) The title of this review focuses on the sensitive cancer therapy, however, only a few part introduce how the ACPs improve the effect of sensitive treatment. Please reorganize the content of review.
2) Comparing with natural ACPs, synthetic peptide with anticancer property has more possibility to be adopted in factory. However, few of parts introduce it. Please add this part which will help readers know completely about ACPs.
3) The English writing should be polished.
Author Response
Thank you for your comments on our manuscripts. I hereby upload the answers to the comments.
Many thanks.

Reviewer 3 Report
Bakare et al. described the relevance of anticancer peptides to treat tumors, their identification methods, delivery systems, mechanisms of action and limitations. They also summarized the current and most common anticancer peptides, mechanisms of action, and chemical sequence in a very informative and practical table, giving the reader the appropriate tools to look for relevant information.
Minor points.
- The authors might modify the manuscript title according to the content of the review article. The current title suggests the manuscript focuses on identifying new anticancer peptides and providing evidence of their biological relevance as it is typically shown in original manuscripts and not in review articles.
- Figures 2, 3, and 4. Could the authors represent the negative or positive charges in the peptide and tumor cell membrane before and after peptide interaction?
- Figure 5. The authors represent the antitumor peptides with irregular blue lines. However, figure 5A shows the peptide (blue), target (ruby-color-like), and a large blue shape. Could the author specify the meaning of each shape? It is not clear if the large blue shape represents a protein. Could the author use the same color for the target protein in B, C, and D? Figure 5C: if the yellow circles represent the cell membrane, could the author write down an abbreviation close to the figure and describe it in the figure legend?
- The authors might include some lines or a paragraph describing if the anticancer mechanism of peptides is different in solid tumors and emphasize its effect on angiogenesis and apoptosis.
Author Response

(The authors gave the same response as above.)

Round 2
Reviewer 2 Report
The authors have made enough modifications.
Author Response
Thank you for your feedback. I hereby upload the answers to the comments as follows.
